# Enhancing Low-Flow Forecasts: A Multi-Model Approach for Rainfall–Runoff Models

**Cynthia Andraos**

Regional Center for Water and Environment, Faculty of Engineering, Saint Joseph University of Beirut, Beirut, Lebanon; cynthia.andraos2@usj.edu.lb

**Abstract:** The expected change in rainfall patterns and the increase in evapotranspiration due to climate change leads to earlier droughts, which aggravate water shortages. To ensure the sustainable management of water resources in these conditions, it is necessary to forecast their evolution. The use of hydrological models is essential for monitoring the water crisis. The conceptual hydrological models used in this study are MEDOR, GR4J, and HBV. They are applied in the Nahr Ibrahim watershed, which is a typical Lebanese Mediterranean basin. While these models simplify complex natural systems, concerns persist about their reliability in addressing drought challenges. In order to reduce the uncertainties, this study develops new robust methods that can improve model simulations. First, a particular series concerning low flows is constructed with the use of hydrological low-flow indices. The multi-model approach is utilized to reach a more accurate unique series while combining the low-flow series generated from the models. This combination is accomplished by using the simple average method, weighted average, artificial neural networks, and genetic algorithms. Better results are generated with the use of these methods. Accordingly, this study led to an improvement in model performances while increasing the reliability of low-flow forecasts.

**Keywords:** rainfall–runoff models; low flows; uncertainties; Mediterranean catchments; multi-models

---

## 1. Introduction

In recent times, there has been a substantial increase in addressing the concept of the Anthropocene as a popular scientific term to designate the influence of human activities on the Earth's system [1,2]. One anthropogenic source is climate change, putting the planet into an emergency state, where impacts on the water cycle are accelerating the crisis [3]. The climate, characterized by fluctuations in precipitation and evapotranspiration, constitutes the foremost factor influencing river flows, thereby leading to low water levels [4,5]. Therefore, any anticipated change in climatic conditions (a decrease in precipitation, alteration in their pattern, and increase in evapotranspiration) impacts the hydrological regime and, more specifically, water availability [6]. Climate change is a key contributing factor to alterations in the low-flow hydrology regime alongside other influential factors.

Numerous studies on the Mediterranean basin agree that the Mediterranean region is one of the most vulnerable areas to climate change [7–9]. The hydrological regime of Nahr Ibrahim, which is a typical Lebanese coastal watershed, is affected by this change. The recession in water sources occurs 15 days to a month earlier. As a result, the dry period is prolonged by this duration. Snow is often replaced by rain floods, and extreme discharge occurs two months earlier [10].

Sustainable water resources planning, and the restoration of ecosystems depend on low-flow extremes and their consequent impacts [11]. In such circumstances, it is necessary to forecast available water quantities during low-flow periods in order to implement appropriate management measures [12,13]. Reliable predictions are key to improving strategic planning with the increase in extreme events [14]. The primary tool for such predictions is hydrological modelling, which enables the simulation of available water

---

quantities and the development of solutions for sustainable water management. In this study, the conceptual model type is chosen with a simplified structure aiming to describe the complete hydrological cycle of a watershed as accurately as possible.

However, in hydrological forecasting, hydrological models generally do not represent reality. This is mainly due to the fact that they greatly simplify the complex natural process of the hydrological cycle. This can lead to inherent uncertainties in the simulations of these models, potentially impacting the accuracy of discharge predictions [15,16] and consequently influencing the values associated with low-flow conditions. The sources of these uncertainties primarily stem from the model structures, parameters, and input data [17]. To enhance the understanding of processes and facilitate more informed decision making, the analysis of systematic uncertainties is indispensable. Various techniques have been developed to reduce these uncertainties, distinguishing between methods that estimate global uncertainties in an aggregated manner and methods that separately consider uncertainty based on the primary source to which it mainly pertains [18,19].

Previous studies have consistently indicated that the model structure is the highest predictive uncertainty contributor [20–22]. This prevalence of model structure uncertainty, among other sources, prompted us to specifically address and evaluate it within this study. A common way to assess uncertainties related to model structures is to use the multi-model approach to improve their performance [23]. This approach uses the outputs of each individual model and combines them to obtain a single result that merges the information contained in each model and compensates for the deficiencies of each other [24–27]. Furthermore, diverse research studies have substantiated the superior performance of multi-model approaches in comparison to a single model employing varied techniques [28–30].

The most frequently employed techniques in multi-model procedures include approaches such as calculating the average or weighted average of the individual model [26,31,32], linear regression [33,34], Bayesian model averaging, which assigns a larger weight to a probabilistic likelihood measure and is one of the most popular techniques used in hydrology [35–37]. Ensemble methods involving the integration of multiple model runs or scenarios to generate a range of possible outcomes are also used [38].

A common critique of multi-model uncertainty analysis using the above methods is its challenge in identifying and incorporating the complete range of possible model structures. Despite these methods having the capability to accommodate numerous structures, the difficulty lies in pinpointing and exploring the entire range of structural possibilities, which can lead to time-consuming and intricate analyses. Additionally, the predictions they provide are sensitive to the specific ensemble members chosen. Considering these limitations, recent studies have introduced the use of artificial intelligence techniques as an alternative approach, especially artificial neural networks [39–42].

Considering the potential advantage of the application of multi-model techniques to hydrological modeling and the lack of studies in the context of low flows, this study aims to analyze the contribution of multi-model combination methods in low-flow forecasting to improve the performance of hydrological models. Three conceptual models (MEDOR, GR4J, and HBV) are selected to generate forecasts for the Nahr Ibrahim watershed in Lebanon. Then, we propose to adapt the simple averages and weighted averages to identify optimal multi-model results that enhance the prediction of the individual hydrological models. These techniques undergo further enhancement within the context of the low-flow scenario. Subsequently, we proceeded to introduce artificial intelligence techniques, specifically artificial neural networks and genetic algorithms. Notably, this marks the inaugural application of these techniques within the low-flow forecast. The purpose of this study is assessed based on the following: (1) the effectiveness of each combinatorial approach compared to single-model results (to use or not use?), (2) comparing the performance of the four methods against each other, and (3) the determination of the most suitable method to use for predicting low-flow scenarios.

The present paper is organized as follows: Sections 2.1–2.3 describe the study area, the available data, and the hydrological models. Then, in Sections 2.4 and 2.5, we elucidate

the methodology and explain the multi-model approaches employed within this study. Section 2.6 illustrates the formulation of low-flow indices and the selection of the series. In Section 3, we discuss the four methods and their outcomes. Section 4 analyses and discusses the results obtained with a comparison between all the methods, and finally, in Section 5, the conclusions of this work are provided.

## 2. Materials and Methods

### 2.1. Study Area

Nahr Ibrahim exemplifies the characteristics of a typical coastal river in Lebanon. It has a total length of 30 km, starting from Mount Mnaitara and flowing into its mouth on the Mediterranean coast. In addition to several streams, it is primarily fed by the following two karstic sources: Afqa (at 1200 m) and Roueiss (at 1265 m). The hydrological regime of the Nahr Ibrahim watershed is characterized by high flows during winter, which is caused by intense precipitation, as well as a sustained flow during spring supported by the melting of the snowpack. The low-flow period begins in May and extends until October. The total basin area is 329 km$^2$ (Figure 1).

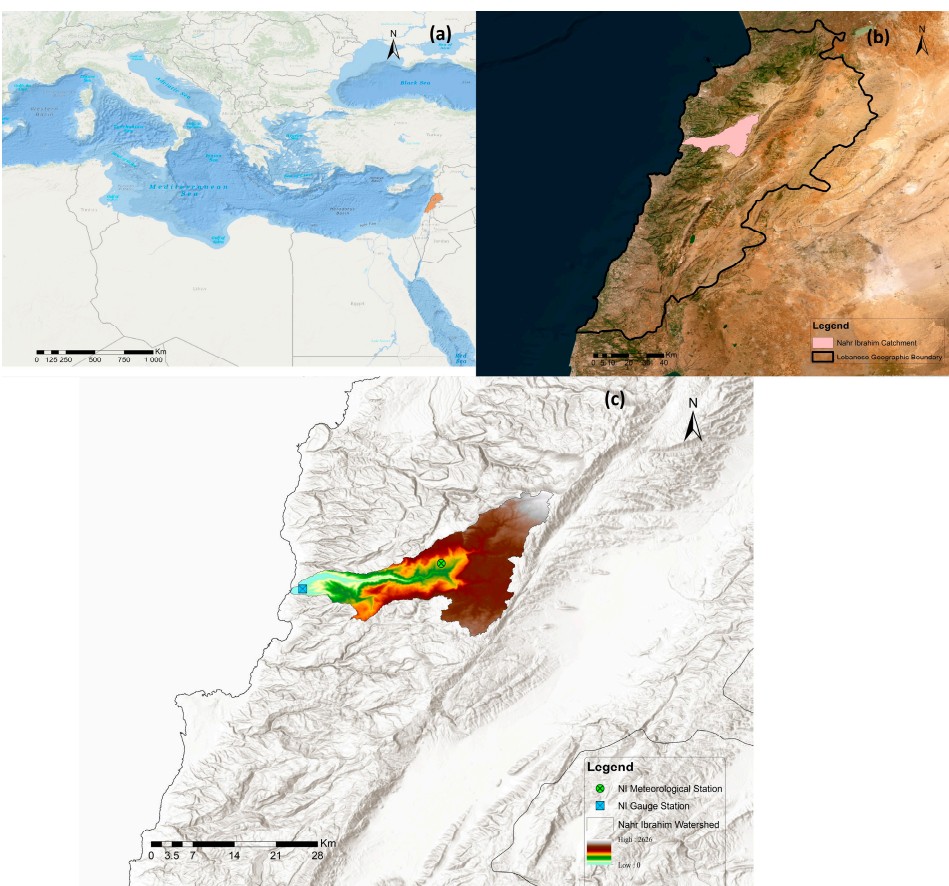

**Figure 1.** (**a**) Location of Lebanon within the Mediterranean region, (**b**) Nahr Ibrahim catchment location, and (**c**) topography of Nahr Ibrahim and hydro-climatic stations.

### 2.2. Hydro-Climatic Data

The climate data used consist of observed daily precipitation and temperature measurements over 17 years (Figure 1).

Initially, the median altitude of Nahr Ibrahim is estimated using the area–elevation curve. In order to provide a more robust representation of the entire watershed, the precipitation and temperature data are spatially averaged at a median altitude, employing temperature and precipitation gradients.

The potential evapotranspiration is estimated using the Hargreaves formula [43], which is the most commonly used formula in Lebanon. The outflow discharge is considered at the river mouth.

Daily data records are utilized to apply the methods described in the following sections. These data undergo rigorous verification to minimize any additional uncertainty associated with the observed data.

### 2.3. Hydrological Models

The three hydrological models (MEDOR, GR4J, and HBV) are lumped models considering the catchment as a single entity with conceptual schemes and components for transforming rainfall into runoff through various mechanisms such as infiltration, surface runoff, subsurface flow, evapotranspiration, storage and flow paths. All these components are represented within reservoirs and parameters as seen in Figure 2.

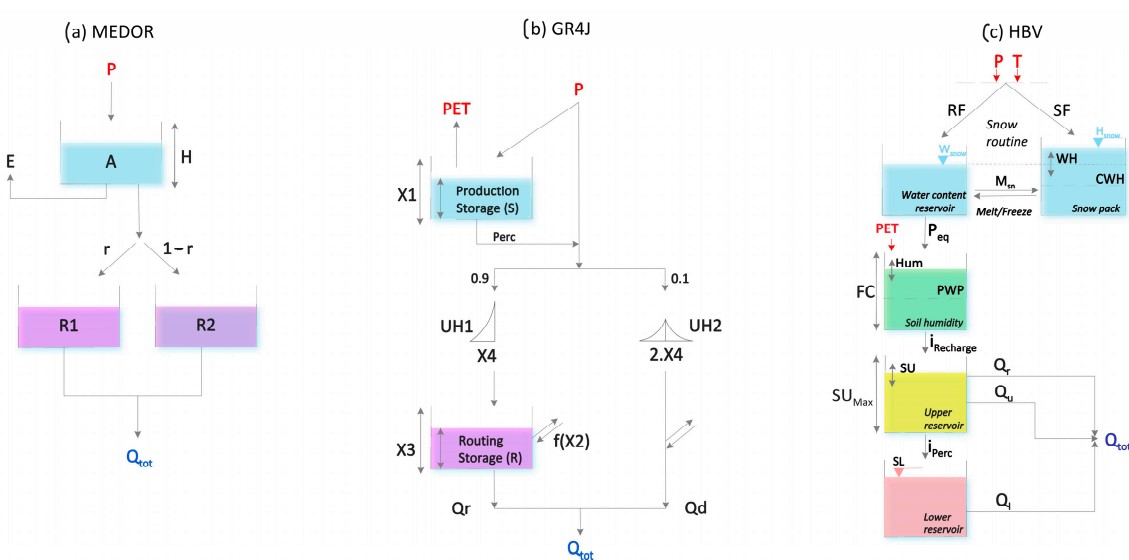

**Figure 2.** Schematic representation of the structures of the three models. (**a**) MEDOR, (**b**) GR4J, and (**c**) HBV.

The chosen models are all applied at a daily time step, using the same inputs of precipitation and potential evapotranspiration, and are calibrated with the same optimization procedure using the Nash–Sutcliffe criteria [44]. A brief description of these three models is presented below:

#### 2.3.1. MEDOR

The MEDOR model ("Méditerranée Orientale") is a conceptual model that has been developed for use in small basins (100 to 1000 km$^2$) under the specific conditions of the Mediterranean environment. This model is represented by two separate functional modules as follows: a production module and a transfer module, each with two parameters [45]. It offers advantages in calibration, needing only concurrent rainfall and discharge data over several non-consecutive years, which is crucial for regions where measurement interruption occurs frequently. The model's production function involves a production reservoir, where rainfall input undergoes a transformation into net basin rainfall and losses, which are regulated by a state variable representing basin moisture. Key to this function are parameters H, the representative reservoir capacity, and EVL, determining losses. The transfer function, encompassing a series of conceptual reservoirs, modulates rainfall distribution to discharge via filters adapted to basin characteristics, with parameters r and T defining transfer and temporal characteristics. MEDOR's mathematical formulation is based on differential equations, which are dependent on the four parameters and require calibration and validation datasets for robust performance evaluation.

### 2.3.2. GR4J

The conceptual model GR4J ("Génie Rural à 4 paramètres Journalier"), with only four parameters, has been widely used in hundreds of watersheds worldwide, covering a wide range of climatic conditions from tropical to temperate and semi-arid watersheds [46]. This conceptual model incorporates elements of reservoir models, featuring a continuous moisture monitoring mechanism to account for antecedent conditions. Structurally, it comprises a production reservoir and a routing reservoir, coupled with unit hydrographs and an external exchange function, to simulate basin hydrological behavior. The mathematical formulation of GR4J involves intricate operations, including rainfall interception, effective rainfall computation, percolation, routing, and exchange processes, with each operation governed by specific parameters. Notably, X1 represents the capacity of the production reservoir, X2 signifies the coefficient of underground exchanges, X3 denotes the one-day capacity of the routing reservoir, and X4 corresponds to the base time of the unit hydrograph.

### 2.3.3. HBV

The rainfall–runoff model HBV ("Hydrologiska Byråns Vattenbalansavdelning") consists of a snow function, a moisture reservoir, and two soil storage reservoirs (upper and lower) [47]. The daily conceptual model HBV was developed by the Swedish Meteorological and Hydrological Institute in the early 1970s. Model inputs include precipitation, temperature, and potential evapotranspiration, with total discharge being the output. The model structure encompasses various parameters and initial conditions. The version of this model used in this study contains nine parameters. These parameters include snow-related factors (melting factor CFmax, critical water content of the snowpack CWH, threshold temperature of rain/snow TT, threshold temperature for snowmelt TTSM), soil storage capacities (maximum FC, soil wilting point, PWP, upper reservoir water level threshold SUmax), and storage coefficients (surface flow storage, Kr, percolation storage, Kperc). Precipitation is partitioned into snow (SF) and rainfall (RF) based on temperature, with equations determining this separation. Snowmelt is calculated using temperature and melting factors, influencing the water content of the snowpack and influencing the equivalent precipitation. Recharge intensity, influenced by model parameters and actual precipitation, regulates soil moisture. Surface runoff, determined by the upper reservoir's water level and storage coefficients, contributes to total discharge alongside subsurface flow from the lower reservoir. The model output, which is the total discharge, is computed by summing surface runoff, subsurface flow, and baseflow.

A snowmelt module based on the degree-day method is added to the inputs of GR4J and MEDOR, which do not consider the effects of snow in their structures. The HBV model, on the other hand, already includes a snowmelt reservoir in its structure.

### 2.4. Methodology

Initially, the calibration and validation of the three hydrological models are carried out. A series of simulations from each model is considered for the subsequent analysis. Low-flow indices are then selected to construct a specific low-flow series. Through a multi-model approach, these chronicles are combined to obtain a single series that is more accurate than the series from each individual model. Four combination methods are used, which are described in the following sections. The methodology and methods employed in this study for low-flow forecasts were developed based on our previous studies [36], where the techniques were extensively explored and refined. Concisely stated, the procedural sequence encompasses the following stages: (1) the calibration and validation of the three distinct models, (2) the creation of low-flow series through the utilization of designated indices based on model outputs, (3) the combination of these series employing multi-model combination methods. The following methodology is summarized in Figure 3.

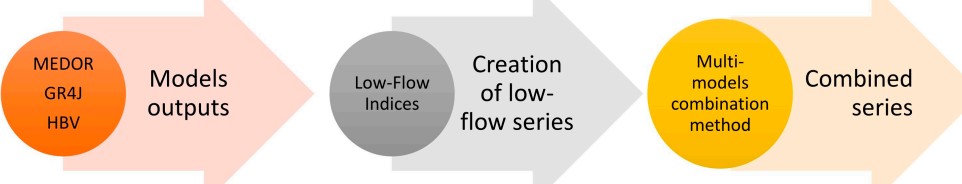

**Figure 3.** Adopted methodology.

### 2.5. Multi-Model Approach

This approach is based on recognizing the imperfection of rainfall–runoff models, and a good description of the uncertainty of the structures can be obtained by using multiple models. It has the following two main objectives: improving the outputs by combining the results obtained from several models and considering the uncertainties linked to the imperfection of the models. The combination of hydrological models makes it possible to use the potential of each model to create a set of simulations that is more representative of the true probability of occurrence. This approach thus provides a single output that expresses the information of several models. Figure 4 illustrates the foundation of the multi-model approach used in this study.

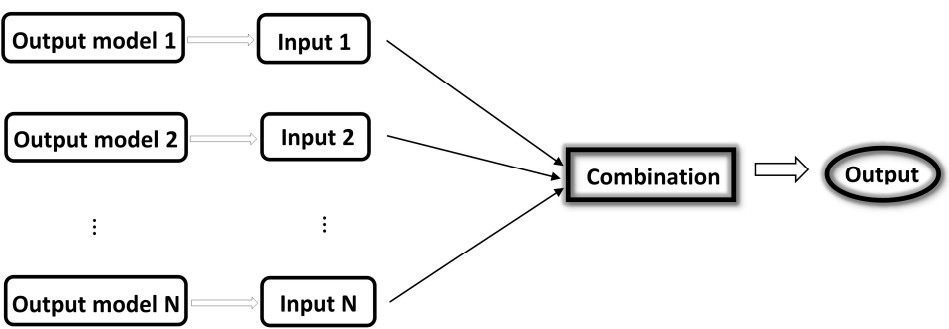

**Figure 4.** Principle of the multi-model approach.

In the context of the multi-model approach, various methods are available to combine hydrological models. In its simplest form, the simulations from different models can be integrated through a straightforward averaging approach, which may also involve certain adjustments using techniques such as moving average and exponential smoothing to refine the final series. Furthermore, more advanced methods that assign weights to the simulations have proven effective. In this study, the determination of the weight assigned to each model is based on the least square method combined with the Lagrange multiplier to satisfy the constraint that the sum of weights equal one [26,32].

The combination of a simulated series is also accomplished by more refined techniques inspired by artificial intelligence. These methods primarily utilize artificial neural networks, which can transform the simulations from the three models into a single series through complex relationships [48]. Then, genetic algorithms [49] used for the same purpose are introduced for the first time in such applications. The following Section 3 delves into these methods, discussing their principles and application to low flows.

### 2.6. Low-Flow Indices and Series

In order to construct a low-flow series, low-flow indices are defined either by fixing a duration or by fixing a flow rate [50]. The selected fixed-duration indices are a flow exceeding 260 days of the year ($Q_{260}$) and a monthly minimum flow ($Q_{\text{min month}}$). As for the selected fixed-flow indices, the flow rate with a threshold of 4 m$^3$/s ($Q_{4\text{m3/s}}$) was chosen. They are opted in a way that represents quantities of interest for low-flow analysis.

Consequently, three low-flow series were constructed by applying the aforementioned indices to the measured series. The output flow values from the hydrological models were considered for each remaining date after eliminating values outside the low-flow period. As a result, the following three low-flow series were generated: (i) a series of flows below $Q_{260}$; (ii) a series of monthly minimum flows; and (iii) a series of flows below 4 m$^3$/s.

## 3. Results

The low-flow series is used to assess the suitability of the four combination methods:

### 3.1. Simple Averages

This is a basic method to combine the outputs of hydrological models. The hypothesis made here suggests that the simple average enables us to take advantage of the strengths of each model. Once the model outputs are estimated, a simple average is calculated. To further explore this trivial combination, moving average and exponential smoothing are introduced to improve the quality of the results. The correlation coefficient, as well as the mean and sum of squared errors (MSE and SSE), are calculated between the initial and combined series to test the credibility of the method (Table 1).

**Table 1.** Results of the combination using simple averaging.

| | GR4J/ Observed | MEDOR/ Observed | HBV/ Observed | Combined Simple/ Observed | Combined with Moving Average/ Observed | Combined with Exponential Smoothing/ Observed |
|---|---|---|---|---|---|---|
| | | | $Q_{260}$ | | | |
| Correlation | 0.26 | 0.38 | 0.32 | 0.35 | 0.40 | 0.39 |
| MSE | 0.81 | 0.55 | 2.15 | 0.56 | 0.41 | 0.30 |
| SSE | 686 | 472 | 1827 | 474 | 347 | 258 |
| | | | $Q_{4m3/s}$ | | | |
| Correlation | 0.42 | 0.64 | 0.53 | 0.58 | 0.66 | 0.43 |
| MSE | 1.79 | 2.07 | 14.18 | 3.06 | 1.62 | 1.26 |
| SSE | 2460 | 2837 | 19,442 | 4193 | 2216 | 1722 |
| | | | $Q_{min\ month}$ | | | |
| Correlation | 0.69 | 0.86 | 0.74 | 0.79 | 0.87 | 0.02 |
| MSE | 249 | 32 | 158 | 107 | 70 | 73 |
| SSE | 23,977 | 3112 | 15,246 | 10,297 | 6710 | 6957 |

The new values of the combined series, even after applying the moving average and exponential smoothing, provided good results but were no better than those of the individual MEDOR model. As such, the resultant values demonstrate favorable outcomes for low-flow series, but they fall short of replacing the superior performance of the best-fit model. Hence, the decision to adopt this method for predicting low-flow scenarios in this regional case was not pursued.

### 3.2. Weighted Averages

When certain models are more accurate than others, it is useful to give them greater influence in the combination procedure. This method assigns more weight to the best models and can be expressed by the following formula:

$$Q_i = \sum_{j=1}^{N} a_j \hat{Q}_{i,j} + e_i$$

where $Q_i$ is the observed flow rate at time instant $i$, $a_j$ is the weight assigned to the $j^{th}$ model that estimates the flow rate $\hat{Q}_{i,j}$, and $e_i$ represents the error.

The complexity of this procedure lies in the estimation of the weights. In this case, the above equation can be viewed as a multiple linear regression model and is solved using the least squares technique, utilizing the Lagrange multiplier to satisfy the constraint of the sum of weights equal to 1 [26]. It means that the weights of each model are determined in a way that minimizes the squared differences between the combined models' output (weighted sum of individual models' outputs) and the measured values. Thus, we obtained a set of weights for each model.

Figure 5 shows that the predicted flows from the combination are closest to the observed flows for the monthly minimum flow series (similarly for $Q_{260}$, $Q_{4m3/s}$, and other periods), confirming the validity of this method in the combination procedure.

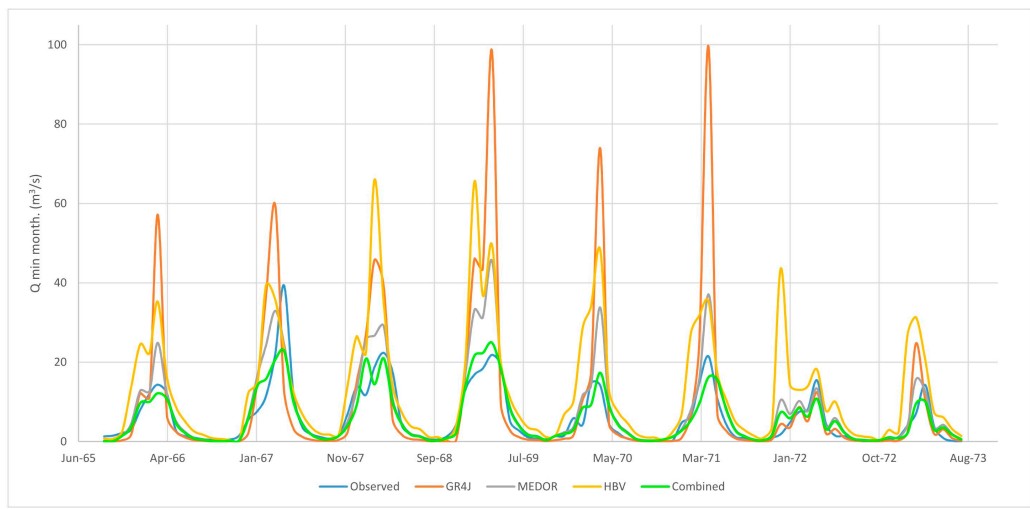

**Figure 5.** Comparison of the observed series, MEDOR, GR4J, HBV outputs, and combined series for $Q_{\text{min month}}$ with the weighted average method.

However, while analyzing this method, it was acknowledged that a significant degree of variability was evident in the weighted values assigned to each model. Notably, the coefficient associated with the MEDOR model exhibited the highest magnitude (close to 1), verifying the outcomes derived from the simple average method, which identified MEDOR as the optimal-fit model in this application. While the results of this weighted average method demonstrated proximity to the measured values (which is a method not to be dismissed), the considerable dispersion in weight values introduced an element of uncertainty concerning the performance of this approach.

In light of this, we decided to investigate more precise methods grounded in artificial intelligence.

### 3.3. Artificial Neural Network

Usually, artificial neural networks (ANNs) are commonly used in hydrology as standalone hydrological models. In this particular application, they were applied in a different perspective. An ANN functions as a computational framework inspired by the human brain's neural structure. Compromising interconnected nodes, or "neurons", organized in layers, an ANN operates by processing input data through these layers and progressively refining its internal weights and biases to optimize its output. In the context of this study, the ANN is trained using the estimated outputs from the three hydrological models during low-flow periods as inputs. Through iterative learning processes, the ANN adapts its internal parameters to generate a transformed output, which, in this case, represents the combined predicted flow rate.

This approach capitalizes on the ANN's capacity to capture complex relationships within data, thereby enhancing the accuracy of final predictions while diverging from the conventional role of ANN as a primary hydrological model.

The utilization of these networks yielded superior results when compared to the measured flow rates. This outcome was examined through correlations between the neural network output series, observed series, and individual model outputs during low-flow periods. The correlation coefficient values are presented in Table 2.

**Table 2.** Correlation coefficients of the combination using neural networks.

| GR4J/ Observed | MEDOR/ Observed | HBV/ Observed | Combined with ANN/Observed |
|:---:|:---:|:---:|:---:|
| | $\mathbf{Q_{260}}$ | | |
| 0.26 | 0.38 | 0.32 | 0.50 |
| | $\mathbf{Q_{4m3/s}}$ | | |
| 0.63 | 0.76 | 0.67 | 0.82 |
| | $\mathbf{Q_{min\ month}}$ | | |
| 0.69 | 0.87 | 0.74 | 0.89 |

Considering the improved hydrological forecasting during low-flow conditions, it is recommended to employ neural networks in multi-model procedures.

As such, the outcomes of this combination method directly support the purpose of this study by showcasing the effectiveness and practical suitability of the ANN's approach in enhancing low-flow predictions within multi-model frameworks.

### 3.4. Genetic Algorithms

Genetic algorithms are optimization algorithms based on techniques derived from genetics and natural evolution. These algorithms mimic the evolutionary process in nature, selecting the best solutions over iterations to reach an optimal outcome. This study employs the standard algorithm [51]. As such, it is used to optimize a set of parameters associated with different types of defined mathematical equations in order to combine the outputs generated by the three different models. The equations used for combining the models' outputs are categorized into the following four types:

Type 1: sum of logarithms: $Q_{comb} = \log a * Q_{GR4J} + \log b * Q_{MEDOR} + \log c * Q_{HBV}$;

Type 2: exponentiation: $Q_{comb} = Q_{GR4J}{}^a + Q_{MEDOR}{}^b + Q_{HBV}{}^c$;

Type 3: fraction: $Q_{comb} = \frac{Q_{GR4J}}{a} + \frac{Q_{MEDOR}}{b} + \frac{Q_{HBV}}{C}$;

Type 4: exponential: $Q_{comb} = e^a * Q_{GR4J} + e^b * Q_{MEDOR} + e^c * Q_{HBV}$.

The values of parameters *a*, *b*, and *c* are optimized for each function using genetic algorithms. This optimization process aims to maximize the alignment between the combined flow rate and the observed low-flow series.

The results reveal that, among the equation types tested, the most suitable equation type for such a combination in the context of low flow is type 1. Therefore, and based on the genetic algorithm's optimization process, the sum of logarithms provides the best fit to predict low flows.

## 4. Discussion

Each of the previously mentioned methods possesses its inherent strengths and limitations. In this discussion, a selection of these attributes that were encountered during the course of this study was highlighted. Then, a comparative analysis was conducted to assess the performance of each approach and determine the most effective combinatorial method.

To begin with, the utilization of the simple average to combine the outputs within a multi-model approach, this method presents on the positive side simplicity, computational efficiency, and a rapid integration capability. However, the equal weighting it assigns to all

models may not account for variations in their individual performance. The vulnerability to outliers, especially for low-flow applications, further characterizes its limitations. In the context of this study, particularly concerning low-flow scenarios, the results obtained through this simple averaging method did not outperform individual predictions of the MEDOR model. Given this outcome, the practical utility of this method for enhancing low-flow predictions is deemed limited and cannot be considered in this specific context.

Subsequently, the weighted averaging method stands as an interesting approach for combining low-flow simulations. It includes enhanced weighting capabilities, allowing for dynamic adjustments to accommodate the strength of each model. However, the subjective weight could introduce biases, demanding careful consideration to avoid compromising outcomes, such as the results we obtained from the high coefficient for the MEDOR model compared to the coefficients of other models. Moreover, the estimation of the weights becomes very expensive in terms of calculation time when the number of models increases. In summation, thorough analysis remains essential to harness the full potential of this method within a multi-model framework.

Following this, the utilization of an ANN presents distinct strengths. It possesses exceptional capability in capturing complex non-linear relationships within data, enabling it to effectively integrate different models' outputs and potentially enhance prediction accuracy. Moreover, ANN can handle intricate interactions between variables that might be overlooked by traditional methods. However, their "black-box" nature makes it challenging to understand the reasoning behind their predictions. In essence, and as per the outcomes of this combination method, it offers the potential to effectively integrate low-flow simulations from different models.

Lastly, the application of genetic algorithms provides a systematic and flexible optimization approach to determine the optimal coefficients and equations to combine the models' outputs. Their ability to explore a wide solution space makes these algorithms suitable for identifying optimal parameters that result in improved predictions. However, a limitation arises from the fact that the success of the genetic algorithms depends on the formulation of the fitness function and can only explore a limited number of equation types due to practical constraints, which could hinder their ability to capture the true underlying relationships. In summary, the results of this combinational method show that it offers a promising avenue for combining low-flow simulations from different models.

On another hand, in order to test the validity of the combination methods and to conclude the robustness of the above techniques, a comparison between the series was carried out. It constitutes one of the fundamental instruments of any scientific approach to confirm the results. We intervene in this section for the purpose of conducting a comparative analysis, aiming to assess the performance of each method in relation to the others. This can be discerned by examining the curves of the observed and predicted cumulative flows. Such a comparative aspect facilitates the quantification of the degree of association and the level of similarity between the compared series. Figure 6 shows an example, among others, of the 1965–1973 series for the $Q_{260}$ flow rates simulated by the GR4J model and those estimated by weighted averaging, the artificial neural network, and the genetic algorithms method. It is evident that the curve generated from GR4J significantly deviates. So, it is reasonable to infer that these three methods exhibit superior performance compared to the individual model outcome. In addition, the cumulative discharge curve resulting from the ANN method is closer to the observed discharge curve in comparison to the curves derived from the weighted average and genetic algorithms. This comparison serves as an illustration of the approach, and similar assessments are conducted with all the cases, showing the same result.

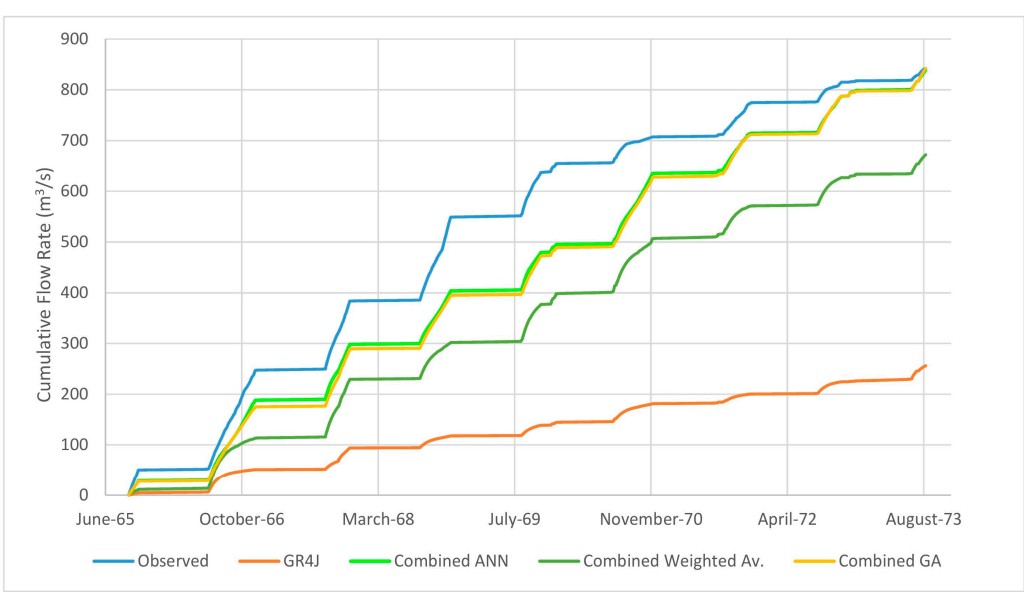

**Figure 6.** Curves of cumulative $Q_{260}$ flows for observed series, simulated by the GR4J model and combination methods.

To sum up, the validation of the combination methods demonstrates their advantages using a single model output, as well as the high accuracy of the ANN method compared to others in low-flow forecast applications.

## 5. Conclusions

This paper examined several procedures for combining simulations from conceptual hydrological models with the aim of improving low-flow discharge forecasting. The proposed multi-model approach utilizes simple averages, weighted averages, neural networks, and genetic algorithms. For each method, a single output combining the three conceptual models was generated. The application was conducted on a Mediterranean watershed.

This study demonstrates the potential for the successful utilization of multi-model methods to enhance hydrological forecasting, particularly during low-flow periods. It also underscores the value of multi-model fusion in hydrological forecasting and opens up avenues for advancing in the field through more modeling techniques based on new technologies. The results have shown the effectiveness and significance of the multi-model methodology proposed in this study compared to other traditional methods. In practical applications, these methodologies could significantly aid water resource management, climate adaptation strategies, and decision making in hydrological engineering projects. The exploration of these opportunities holds the promise of advancing our ability to address the complex challenges posed by low-flow scenarios and contributing to sustainable water resource practices.

**Funding:** This research received no external funding.

**Data Availability Statement:** The data presented in this study are available on request from the corresponding author.

**Acknowledgments:** I would like to express my deepest gratitude and appreciation to the late Wajdi Najem for his invaluable guidance and supervision throughout the course of this research. Sadly, Najem passed away before the finalization of this article, leaving an irreplaceable void in our hearts and academic community.

**Conflicts of Interest:** The author declares no conflicts of interest.

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
