# Peer review of "Enhancing Low-Flow Forecasts: A Multi-Model Approach for Rainfall–Runoff Models"

_hydrology, doi:10.3390/hydrology11030035_

Round 1

Reviewer 1 Report

Comments and Suggestions for Authors

A very good scientific paper concerning one of the major national basin of Lebanon.This study will help the engineers of the Hydraulics Directorate of the Ministry of Water and Energy of Lebanon and the Water Authority of Beirut and Mount Lebanon to seek a better understanding of the river behavior in dry seasons in relation with the future planned infrastructures.

I congratulate the author.

Author Response

Dear Reviewer,

Thank you very much for your kind words and congratulations on the paper. Your recognition means a great deal to me, and I truly appreciate the time you took to review it. Knowing that this study could potentially contribute to the efforts of the Hydraulics Directorate of the Ministry of Water and Energy of Lebanon, as well as the Water Authority of Beirut and Mount Lebanon, is incredibly rewarding. I'm glad to hear that you found value in the research and that it may assist in enhancing our understanding of river behavior during dry seasons, especially in the context of future infrastructure plans.

Once again, thank you for your encouragement and support.

Reviewer 2 Report

Comments and Suggestions for Authors

The study aimed to improve hydrological forecasts by exploring different combination methodologies using three hydrological models. The manuscript is well-written and explains the methodology satisfactorily. However, there are some questions listed in the general comments. This methodology seems promising and has great potential. Therefore, I recommend acceptance with minor revisions. Please see my comments below.

General comments:

1. The introduction needs to provide more context by mentioning other approaches, such as ensemble streamflow/hydrology forecast, and explaining why the proposed methodology is better compared to others.

2. In lines 41-43 of the introduction, the authors incorrectly refer to the hydrological regime associated with seasons as a drought instead of the dry season.

3. Fig. 1 needs improvement. The Lebanon boundary is not clear in Fig. 1b, and I recommend merging Fig. 1c and 1d.

4. The authors need to include the data period for both gauge and meteorological stations.

5. The rationale for choosing Q260 and Q = 4 m3/s needs to be explained.

6. It may be beneficial to combine Figures 3 and 4.

7. The results section needs to be separated from the methodology section.

8. A table with genetic algorithm results needs to be included.

9. In Figure 6, the observed curve is higher than all other curves, and the authors need to double-check the figure and lines 367-370.

10. The authors should consider comparing with the MEDOR model instead of the GR4J model, as the MEDOR model produces better results. Moreover, the authors need to explain why they chose Q260 rather than Qmin month.

11. The authors need to discuss their results in comparison to other approaches that aim to improve hydrological forecasts, as suggested above.

Reviewer 3 Report

Comments and Suggestions for Authors

I would like to congratulate the author, it is a very good effort in the section of hydrology. The manuscript is well-written and robust. The language is the appropriate one. English are fine. 

While reading the article, I have mentioned some points that need further explanation. Firstly, methods should be more clear. It is necessary authors to mention the parameters of the models and give some information for the models , not something extensive, one or two sentences. 

Secondly, in line 213, why authors have chosen the threshold of 4 m3/s? It needs an explanation. Why not a smaller value?

Furthermore, the data that have been used as inputs in the models are in public? Is there any access? Where have they been taken from? The meteorological station? More information are necessary to be presented. 

Last but not least, the biblioraphy needs improvement, more recent references are required for a better justification. 

I propose a major revision of the manuscript.

Round 2

Reviewer 3 Report

Comments and Suggestions for Authors

I have no further questions. The manuscript can be published.